

# *TLR2* polymorphism (rs650082970) is associated with somatic cell count in goat milk

Jernej Ogorevc[1], Mojca Simčič[1], Minja Zorc[1], Monika Škrjanc[2] and Peter Dovč[1]

[1] Biotechnical Faculty, Department of Animal Science, University of Ljubljana, Ljubljana, Slovenia
[2] Faculty of Chemistry and Chemical Technology, University of Ljubljana, Ljubljana, Slovenia

## ABSTRACT

Pathogens invading the mammary gland are recognized through a range of pattern recognition receptors (PRRs), residing on the plasma membrane of mammary epithelial cells. Toll-like receptor 2 (*TLR2*) signalling is responsible for recognition of Gram-positive bacteria, which are the most common mastitis-causing pathogens in goats. Somatic cell counts (SCC) in milk are routinely determined in goat dairy flocks and serve as an indicator of milk quality, which is highly correlated to intramammary infections. Recently, a single nucleotide polymorphism of the *TLR2* was suggested to be associated with SCC in goat milk. To further test the suggested association, we genotyped 61 Slovenian Alpine goats included in the dataset. The effect of the genotype was analysed using the general linear model (GLM) procedure of SAS/STAT software. We found the *TLR2* genotypes significantly ($p = 0.0007$) associated with milk SCC. Animals with the *A/G* genotype had significantly ($p \leq 0.05$) lower SCC value in milk compared to the *G/G* genotype. Our data suggest that the *A* allele is the minor one and is associated with lower milk SCC. In the current study, we provide a validated PCR-RFLP based genotyping assay for the *TLR2* SNP (rs650082970) and confirm its association with milk SCC. Further studies to confirm the association on a larger number of animals of different breeds and to explain functional consequences of the polymorphism in relation to SCC are encouraged.

## INTRODUCTION

Mastitis, an inflammation of mammary tissue, is the major concern in the dairy sector, causing economic loses, animal welfare concerns, and reduced quality of milk and milk products. Mammary epithelial cells are the first barrier against invading pathogens and play a key role in recognition of pathogens and in induction of innate immune response during intramammary infection (*Stelwagen et al., 2009*). The recognition of pathogens by the innate immune system is mediated through pattern recognition receptors (PRRs), which recognise evolutionarily conserved pathogen-associated molecular patterns (PAMPs), present on the surface of pathogens (*Mogensen, 2009*). Toll-like receptors (TLRs) were the first recognized and are the most well-characterized PRRs(*Kawai & Akira, 2011*). A wide range of PRRs, including TLRs, is expressed on plasma membranes of mammary epithelial

Corresponding author
Peter Dovč, peter.dovc@bf.uni-lj.si

cells (*Ezzat Alnakip et al., 2014*). *TLR2* is important for recognition of Gram-positive bacteria (*Schroder et al., 2003*), which are the most common mastitis-causing pathogens in goats (*Bergonier et al., 2003*) and *TLR2* is therefore one of the crucial PRRs responsible for mammary gland immunity in small ruminants.

Somatic cell count (SCC) is a widely used indicator of milk quality and overall udder health status in dairy herds. Mammary infections correlate well with elevated levels of somatic cells in milk in cattle. This relation is a bit less straightforward in goats where SCC are generally much higher (*Paape et al., 2007*) due to the apocrine mechanism of milk excretion in goats, resulting in cytoplasmic particles present in milk, which could be mistakenly counted as somatic cells (*Paape & Capuco, 1997*). However, with proper optimization of the detection methods and use of goat specific SCC standards, SCCs serve also as an indicator of mammary health status in goats and are routinely determined in milk from dairy goat flocks (*Wilson, Stewart & Sears, 1995*; *Raynal-Ljutovac et al., 2007*). Interestingly, no generally accepted grading criteria or legal standards exist for classification of goat milk according to SCC as are established for cow's milk. Some authors (*Silanikove, Merin & Leitner, 2014*) suggest that grade A goat milk should contain up to 840,000 somatic cells/ml, and that goat milk with more than 3,500,000 cells/ml should not be accepted for marketing (*Leitner, Silanikove & Merin, 2008*).

It has been shown in cattle that expression of *TLR2* is induced during intramammary infections (*Goldammer et al., 2004*; *Mitterhuemer et al., 2010*; *Gunther et al., 2011*). In addition, polymorphisms of *TLR2* have been associated with milk SCC (*Bai et al., 2012*). A single nucleotide polymorphism (SNP) rs650082970 was recently proposed to be associated with goat milk SCC, by sequencing the *TLR2* target region of 39 goats of different breeds included in the dataset (*Ruiz-Rodriguez et al., 2017*). To further test the suggested association in goats, we designed a PCR-RFLP based genotyping assay and genotyped 61 does of Slovenian Alpine goat breed from a single dairy farm. Statistical analysis was conducted to determine whether the *TLR2* genotypes are associated with SCC in goat milk.

## MATERIALS & METHODS

### Animals, phenotypic data and DNA extraction

Hair samples from 61 does from Slovenian Alpine goat breed were obtained from animals included in the AT4 milk recording system according to the latest ICAR (https://www.icar.org/) guidelines (subsequent test days take place at 28 to 34 day intervals). A total of 863 AT4 milk records were extracted from the Central Database for Small Ruminants, maintained by the Department of Animal Science (Biotechnical Faculty, University of Ljubljana). Phenotypic data include age, parity, consecutive milk recording, milk composition (fat, protein, lactose, urea) and somatic cell count (somatic cell number per ml) for the period from July 2015 to November 2017. There were seven milk recordings for each doe yearly in 2017 and 2016 and four in 2015. Does were in their first to fourth parity. Genomic DNA was extracted from hair follicles using Isolate II Genomic DNA Kit (Bioline, London, UK) according to the manufacturer's instructions.

The collection of animal samples was carried out in accordance with the recommendations of the European Union Directive 2010/63 and the national animal testing legislation.

## PCR-RFLP analysis and sequencing

Polymerase chain reaction (PCR) was performed to screen for the SNP rs650082970 in the PCR amplified 442 bp fragment of the *TLR2* gene, using forward: 5′-ATCTGCGGACCCT GAAAGTA-3′ and reverse: 5′-GCTGTAAAATCGCCAATTCC-3′ primer pair. The PCR primers were designed in Primer3 primer design tool (http://bioinfo.ut.ee/primer3-0.4.0/), using goat *TLR2* source sequence JQ911706 (GeneBank). The amplification reactions were performed as follows: 5 min at 95 °C, 35 cycles at 95 °C for 30 s, 58 °C for 30 s, and 72 °C for 30 s, followed by final elongation step at 72 °C for 3 min. The reaction volume was 20 $\mu$l and contained 1 × PCR buffer, 0.75 $\mu$M primers, 150 $\mu$M dNTPs, 1.2 mM MgCl$_2$, 0.5 U DNA Taq polymerase (Thermo Fisher Scientific, Waltham, MA, USA), and approximately 50–200 ng of template DNA. The PCR products were digested using restriction endonuclease *Vsp*I (ER0911; Thermo Fisher Scientific, Waltham, MA, USA). The restriction reaction mixture consisted of 10 $\mu$l PCR product, 1.5 $\mu$l restriction buffer, 3.25 $\mu$l H$_2$O, and 0.25 $\mu$l (2.5 U) of *Vsp*I, and was incubated for 3 h at 37 °C. DNA fragments after restriction were analyzed on 2.5% agarose gel stained with ethidium bromide. In the case of *A* allele fragments of 288 and 153 bp were obtained and in the case of G allele the PCR product remained uncut (442 bp). The PCR/RFLP method was confirmed using Sanger sequencing of PCR products representing all three genotypes. The fragments were treated with exonuclease I (ExoI) and alkaline phosphatase (FastAP) (both Thermo Fisher Scientific, Waltham, MA, USA) for 15 min at 37 °C prior to sequencing using Big Dye v3.1 sequencing kit (Thermo Fisher Scientific, USA) and the forward primer. The fragments were purified using EDTA and ethanol, resuspended in formamide and sequenced on ABI3100 gene analyzer (Applied Biosystems, Foster City, CA, USA).

## Statistical analysis

A chi-square test was applied to test the genotype frequencies for deviations from Hardy-Weinberg equilibrium. Forty-five *G/G* and 14 *G/A* animals were included in the analysis. The rare homozygote *(A/A)* group size wass small ($n = 2$), therefore it was excluded from the analysis. The analyzed data for 61 goats represent 863 recordings (14.6 per goat in average). The data was not averaged by any of the variables included in the model (genotype, parity, consecutive milk recording, and SCC). Since SCC has a highly skewed distribution, the data was log-transformed to obtain a normal distribution. Analysis of variance was performed with the general linear model (GLM) procedure of the SAS software (SAS Institute Inc. 2001, USA) according to the following model:

$$y_{ijk} = \mu + J_i + K_j + G_k + e_{ijk}$$

where $y_{ijk}$ is log-transformed value of the SCC, $\mu$ is overall mean, $J_i$ is fixed effect of parity ($i = 1$–4), $K_j$ is fixed effect of consecutive milk recording ($j = 1$–7) (days in milk divided by recording interval), $G_k$ is the effect of the genotype (k = *G/A*, *G/G*), and $e_{ijk}$ is residual error. Statistical significance was declared at $p \leq 0.05$.

## RESULTS

Genotyping of the 61 samples for the *TLR2* rs650082970 SNP revealed 45 *G/G*, 14 *G/A* and 2 *A/A* genotypes, which corresponds to the allele frequencies of 0.85 and 0.15 for alleles *G* and *A*, respectively. The observed genotype frequencies do not deviate from Hardy-Weinberg equilibrium according to the chi-square test ($p > 0.05$). The *G/G* and *G/A* frequencies are similar to the previously reported results (*Ruiz-Rodriguez et al., 2017*), but in our sample collection we also found two animals with the scarce *A/A* genotype (due to low number we did not include them in the statistical analysis). Additionally, from publicly available database Genome Variation Map (http://bigd.big.ac.cn/gvm/) (*Song et al., 2018*) we extracted goat variation data for 211 goats of different populations (breeds) and estimated the average frequencies of the *G* and *A* alleles to approximately 0.8 and 0.2, respectively. Allele frequencies seem to differ across breeds/populations with *A* allele frequency ranging from 0 to 0.30.

The validity of the PCR-RFLP genotyping method was confirmed by Sanger sequencing of PCR products for all the three different genotypes. The nucleotide sequences of PCR products matched the *TLR2* target sequence (RefSeq NM_001285603.1) and are in accordance with the genotypes determined by the PCR-RFLP method (Figs. 1A, 1B).

In the analysis of variance fixed effects of parity ($p = 0.0088$), consecutive milk recording ($p < 0.0001$) and genotype ($p = 0.0007$) were included. Generally, SCC was the lowest at the first milk recording and as expected increased with consecutive milk recordings (days in milk). Similarly, mean SCC was the lowest in the first parity and increased with consecutive parities. The statistical analysis showed significant effect of the *TLR2* genotype on the SCC. SCCs in milk were lower in heterozygotes compared to *G/G* homozygotes (Table 1).

## DISCUSSION

Heterozygosity could be an advantage in pathogen recognition (*Lenz et al., 2013*), but in our case there were not enough *A/A* goats ($n = 2$) to assess the effect of the heterozygote advantage. However, despite there were only two *A/A* animals it should be noted that their average SCC value was lower than the average SCC values of the other two genotypes (not implying overdominance). From the available data it seems that the *A* allele is the minor allele in goat populations and associated with lower SCC in goat milk, which makes the marker interesting for implementation to selection schemes for increased mastitis resistance. There is however some controversy whether selection to low SCC could eventually make animals more prone to mastitis, but the research shows that this is not the case (for a review see *Rainard et al., 2018*).

The analysed *TLR2* polymorphism does not appear to have an obvious effect on the structure of the protein (*Ruiz-Rodriguez et al., 2017*), therefore it is possible that the analysed SNP is not the causative polymorphism associated with SCC in goat milk, but could be in linkage disequilibrium with the actual causative allele(s). Further studies are required to pile up the evidence for the SNP-SCC association in different breeds and to explain possible physiological effects of the polymorphism in relation to SCC and mastitis resistance.
A)

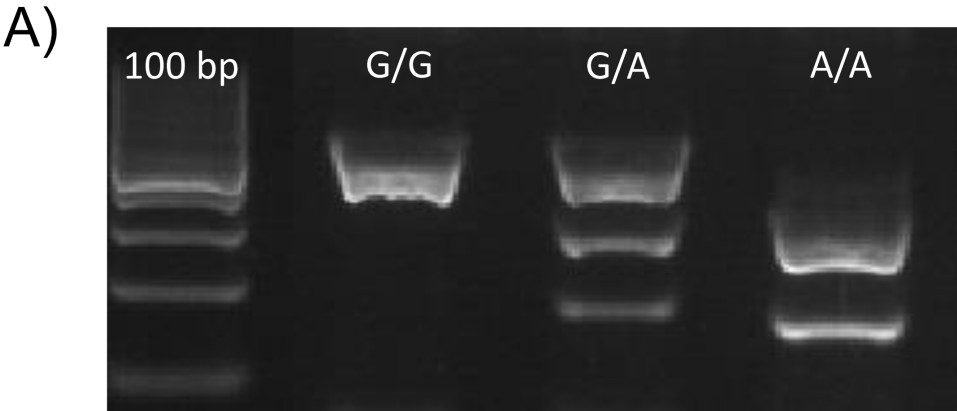

B)

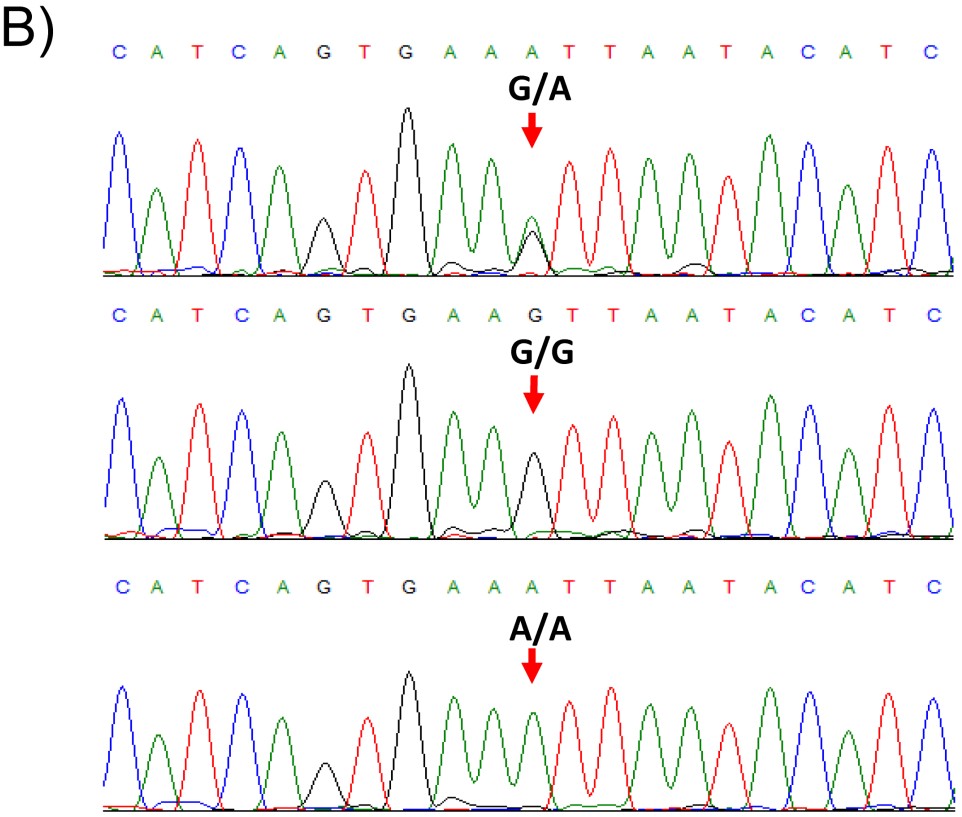

**Figure 1** **Genotypes of the TLR2 locus for the SNP (rs650082970).** (A) Gel electrophoresis of the PCR-RFLP assay for genotyping the rs650082970 SNP. Genotypes *G/G, G/A* and *A/A* are presented. (B) Sanger sequences of the PCR products for the three genotypes. Two peaks are clearly visible in the case of heterozygote *(G/A)* animal.

**Table 1  SCC ($\times 10^3$) by the genotype of the TLR2 polymorphism (rs650082970).**

| Genotype | Goats | Records | SCC ($\times 10^3$) | | |
| --- | --- | --- | --- | --- | --- |
| | | | Mean | Log *LSM | *SE |
| A/A | 2 | 35 | 577.11 | / | / |
| G/A | 14 | 208 | 1055.38 | 6.58[c] | 0.16 |
| G/G | 45 | 620 | 1475.66 | 6.89[d] | 0.15 |

Notes.

*LSM, least squares means; SE, standard error.

[c,d] superscript letters denote statistically significant differences among groups of animals carrying different genotypes.

## CONCLUSIONS

In this study, we provide a validated PCR-RFLP based genotyping assay for the *TLR2* SNP (rs650082970) and confirm the association of this SNP with milk SCC on a sample of the Slovenian Alpine breed does. Animals with the *A/G* genotype had significantly ($p \leq 0.05$) lower SCC in milk compared to the *G/G* genotype. The *A* allele seems to be the minor allele in goat populations and associated with lower milk SCC. Further studies are required to confirm the SNP-SCC association on a large number of animals in different breeds and to explain possible physiological effects of the polymorphism in relation to SCC and mastitis resistance or to identify the actual causative nucleotide(s), possibly linked with the analysed SNP.

## ACKNOWLEDGEMENTS

The authors would like to thank Dr. Tine Pokorn and Dušan Birtič for technical assistance.

### Funding

This work was supported by the Slovenian Research Agency (grant number J4-7328) and by the Slovenian Research Agency and Ministry of Agriculture, Forestry and Food (grant number V4-1416). The funders had no role in study design, data collection and analysis, decision to publish, or preparation of the manuscript.

### Grant Disclosures

The following grant information was disclosed by the authors:
Slovenian Research Agency: J4-7328.
Slovenian Research Agency and Ministry of Agriculture, Forestry and Food: V4-1416.

### Competing Interests

The authors declare there are no competing interests.

### Author Contributions

- Jernej Ogorevc conceived and designed the experiments, performed the experiments, prepared figures and/or tables, authored or reviewed drafts of the paper, approved the final draft.

- Mojca Simčič analyzed the data, contributed reagents/materials/analysis tools, authored or reviewed drafts of the paper.
- Minja Zorc conceived and designed the experiments, analyzed the data, prepared figures and/or tables, authored or reviewed drafts of the paper.
- Monika Škrjanc performed the experiments, sample collection.
- Peter Dovč conceived and designed the experiments, contributed reagents/materials/-analysis tools, authored or reviewed drafts of the paper, approved the final draft.

### Animal Ethics

The following information was supplied relating to ethical approvals (i.e., approving body and any reference numbers):

Hair samples were obtained during routine hair sampling for the paternity testing recommended by the breeding program.

### Data Availability

Raw data is available as a Supplemental File.

### Supplemental Information

Supplemental information for this article can be found online at http://dx.doi.org/10.7717/peerj.7340#supplemental-information.

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
