# Peer review of "TLR2 polymorphism (rs650082970) is associated with somatic cell count in goat milk"

_PeerJ, doi:10.7717/peerj.7340_

## Round 0.1 · original submission · Minor Revisions

Given the comments provided by the reveiwers, minor revisions are required. Please follow their recommendations before sending your revised document.

·

Basic reporting

The paper deals with genetic polymorphism and association to goat milk quality, namely SCC. It is professionally made. The approach of authors is usual in last years and does not bring any novelty.

Experimental design

The design follows usual standards for such analyses. The number of genotyped goats is relatively low, please emphasize it in the Conclusions section. What is the source of primers? Give the reference, or write that you have designed them, give GenBank number, program for design etc.

Validity of the findings

The validity is in relation to the size of the group in analysis. But the research was made on the good professional level, brings reliable results and could be the basis for next studies. Which parity was the best? The A allele was found to be minor, but it is associated with lower SCC. Try to write some short reflection of the fact.

Additional comments

Please revise the paper as suggested above.
Some formal notices:
Abbreviations of genes, genotypes and alleles must be written in italic, revise it throughout the text.
Part C of the Fig. 1 is not necessary, but this notice is not crucial.
R. 102, T allele?

Reviewer 2 ·

Basic reporting

Well written, only minor changes suggested below to improve writing.

Experimental design

Fine; no issues with any of the criteria listed as standards. A couple of minor concerns regarding figure 1, as indicated in comments to the authors

Validity of the findings

Strong impact since it is only the second study to find the association reported, which may be of benefit for improvement of dairy production.

Additional comments

Methods: clarify whether SCC data was first averaged per goat, and then averaged per genotype; or alternatively whether the SCC dataset was not averaged per goat, and only separated by genotype regardless of number of recordings per goat.

Do the authors want to test association by genotype for maximum SCC values?

Line 149: N=2 is too low a number to support or not support any hypothesis, so this statement needs to be greatly qualified, e.g., "There were not enough A/A goats to examine the possibility of heterozygote advantage, although it is notable the average SCC value for the two A/A goats was lower than the average for G/A heterozygotes or G/G homozygotes."

Figure 1A: If GG has one band, and AA has two bands, then why are there not 3 bands for GA?

Figure 1C: There is great overlap in the 95% confidence intervals for the two genotypes. How is this consistent with a significant difference? Is there a better way to plot this to indicate the difference is significant?

Minor comments, by line
5-Indicate correspondent with an asterisk
18-"pathogens invading THE mammary gland"
Throughout: "Slovenian Alpine goats" might be best written as "Alpine breed Slovenian goats"
25-"in the milk recording" should be "in the dataset"
Throughout: Gram positive should be capitalized
37-"one of the major issues" should be "is of major concern"
41"pathogens in" should be "pathogens by the"
45 "precisely" should be "well"
47 "represent" should be "are"
48 "and is" should be ". TLR2 is"
60 "known for cow" should be "established for cow's"
67 remove "the"
70 "milk recording" should be "the dataset"
92-93 Are the primer pair original or previously published? Please specify.
127, if not significant p should be greater than, not less than 0.05
142 expectedly should be "as expected"
149"the lowest SCC in the milk of goats with A/A genotypes" should be "the SCC in the milk of goats with A/A genotypes was lower than that of the other genotypes"
149 N=2 is simply too low a number to support or not
172 "Dr." should be capitalized
References: all taxonomic names (Capra hircus, E. coli) should be italicized
Table 1 header: according to the genotype" should be "by the genotype of TLR2 (rs65...

---

## Round 0.2 · accepted · Accept

I apologize for the delay in the decision on the MS. The previous AE had become unresponsive but I have stepped in to render the decision. Both reviewers are satisfied with your revisions and the MS appears to be well written and rigorous.

·

Basic reporting

The paper is made on good professional level.

Experimental design

Good, usual in the research of this kind.

Validity of the findings

The paper brings new knowledge in the field.

Additional comments

The authors revised the manuscript, and I agree with publishing.
R. 105, G allele must be written in italic.

Reviewer 2 ·

Basic reporting

good

Experimental design

good

Validity of the findings

good